# The association between the risk perceptions of COVID-19, trust in the government, political ideologies, and socio-demographic factors: A year-long cross-sectional study in South Korea

Yo Han Lee[1☯], Hyun-Hee Heo[2☯], Hyerim Noh[3], Deok Hyun Jang[4], Young-Geun Choi[5*‡], Won Mo Jang[6,7*‡], Jin Yong Lee[7,8]

1 Department of Preventive Medicine, Korea University College of Medicine, Seoul, Republic of Korea, 2 Institute for Future Public Health, Graduate School of Public Health, Korea University, Seoul, Republic of Korea, 3 Department of Public Health Sciences, Graduate School of Public Health, Seoul National University, Seoul, Republic of Korea, 4 Research Analytics & Communications, Gallup Korea, Seoul, Republic of Korea, 5 Department of Statistics, Sookmyung Women's University, Seoul, Republic of Korea, 6 Department of Public Health and Community Medicine, Seoul Metropolitan Government – Seoul National University Boramae Medical Center, Seoul, Republic of Korea, 7 Department of Health Policy and Management, Seoul National University College of Medicine, Seoul, Republic of Korea, 8 Public Healthcare Center, Seoul National University Hospital, Seoul, Republic of Korea

☯ These authors contributed equally to this work and as co-first authors.
‡ YGC and WMJ also contributed equally to this work and as co-corresponding authors.
* ygchoi@sookmyung.ac.kr (YGC); thomasj@snu.ac.kr (WMJ)

Data Availability Statement: Data from this study cannot be publicly shared, because we have used third-party data from Gallup Korea, and are not

## Abstract

Risk perception research, targeting the general public, necessitates the study of the multi-faceted aspects of perceived risk through a holistic approach. This study aimed to investigate the association between the two dimensions of risk perception of COVID-19, i.e., risk as a feeling and analysis, trust in the current government, political ideologies, and socio-demographic factors in South Korea. This study used a year-long repeated cross-sectional design, in which a national sample (n = 23,018) participated in 23 consecutive telephone surveys from February 2020 to February 2021. Most factors differed in the magnitude and direction of their relationships with the two dimensions of risk perception. However, trust in the current government, alone, delineated an association in the same direction for both dimensions, i.e., those with a lower level of trust exhibited higher levels of cognitive and affective risk perception. Although these results did not change significantly during the one-year observation period, they are related to the political interpretation of risk. This study revealed that affective and cognitive risk perceptions addressed different dimensions of risk perception. These findings could help governments and health authorities better understand the nature and mechanisms of public risk perception when implementing countermeasures and policies in response to the COVID-19 pandemic and other public health emergencies.

entitled to share the data. Gallup Korea Daily Opinion survey is a telephone research program that has been operated weekly by Gallup Korea since January 2012. It examines basic state-run indicators, including presidential job performance evaluation and political party support, and Koreans' thoughts on major political issues, economy, society, life and culture. Results of basic analysis will be released every Friday at 10 a.m. on Gallup Korea's website (www.gallup.co.kr). Gallup Korea plans and pays for itself, and anyone interested can use the results of the survey for free. However, the use of raw data from the Gallup Korea is allowed only for researchers conducting a joint study with a Gallup Korea researcher. Detailed data approval procedures are carried out in accordance with Gallup Korea's internal guidelines. More information about sharing the data can be obtained by contacting press@gallup.co.kr.

**Funding:** This research received funding from Catholic University of Korea (K2225791), National Research Foundation of Korea (NRF-2022S1A5B5A16057001), and National Research Foundation of Korea (2020R1G1A1A01006229). However, any funder did not play any role in the study design, data collection, analysis, interpretation, and publication decision.

**Competing interests:** Jang DH is affiliated with Gallup Korea (https://www.gallup.co.kr/), but did not receive any funding from them for this. This does not alter our adherence to PLOS ONE policies on sharing data and materials.

## Introduction

Tracking the trajectory of people's risk perception and related factors is crucial for an improved understanding of health-related behavior. According to the decision theory of health behavior, people who perceive a greater risk of disease, illness, or injury are generally more motivated to practice preventive and protective behaviors [1, 2]. Thus, understanding risk perception in the context of a pandemic could play a crucial role in risk management, such as social distancing and vaccination compliance. Furthermore, risk perception of emerging infectious diseases may affect some particular behaviors and a set of behavioral patterns called the behavioral immune system [3, 4]. The behavioral immune system includes the detection of and response to potential pathogens using psychological and behavioral defense mechanisms. In addition, risk perception was associated with pharmaceutical and non-pharmaceutical preventive behaviors during the COVID-19 pandemic [5–10].

Previous research has shown how humans perceive risk could be conceptualized into two main dimensions—cognitive and affective [11–13]. The cognitive dimension (risk as an analysis), in which risk is analytically regarded, represents the logical, rational, and scientific consideration of risk management. It relates to how people come to know and understand risks. However, the risk is viewed emotionally within the affective dimension (risk as a feeling). It represents a quick, instinctive, and intuitive reaction to risks, and it refers to how individuals feel about risks. Affective risk perception can be understood as a type of heuristics (mental shortcut) process, which is more predictable than cognitive risk perception [14–19]. Several studies have revealed that affective risk perception of infectious disease was related to health behaviors [4, 7–9, 20–22]. Furthermore, longitudinal studies in China revealed that worry and anxiety decreased after the peak of the epidemic [23]. However, to the best of our knowledge, no research has investigated the changes in the two dimensions of risk perception and its associated factors concerning the pandemic and its progress.

Risk perceptions can be influenced by multiple factors, such as socio-demographic, psychological, and politico-contextual variables, across time [24]. In previous studies, trust in institutions was found to be heuristic, which was associated with risk perception [25, 26]. The higher the trust in the government, the lower the level of risk perception of emerging infectious diseases during an outbreak [27–29]. Furthermore, political ideologies may be related to the risk perception of COVID-19 [30–32]. However, this association varied in different contexts. For example, conservatives, more or less, were likely to perceive COVID-19 risks according to the political context. Risk perceptions, perceived susceptibility, and risk severity often relate to the media representation of a threat or risk [33, 34]. A systematic review of news coverage related to the H1N1 pandemic outbreak revealed that enormous volume of news, overemphasis on threat protection, and news coverage with a frightening tone and manner influenced the amplification of the perceived risk [35]. In addition, a meta-analysis of 47 studies suggested that concern regarding COVID-19 infection was associated with media exposure worldwide [36].

Concerning the socio-demographic factors, older people perceived a lower risk of contracting COVID-19, yet a higher risk of dying due to COVID-19 [37, 38]. The risk level was higher for women and those with higher economic status and education [39–45]. However, in previous risk perception studies, age, gender, economic status, and education, as socio-demographic variables, had a weak or non-significant association with risk perception [15].

Until February 2021, the pandemic could be divided into five phases according to the upsurge of confirmed cases in South Korea [46]. Three cluster outbreaks related to religious facilities, large-scale downtown gatherings, nursing homes, and healthcare facilities occurred between January 2020 and February 2021. The five phases were as follows: before the first

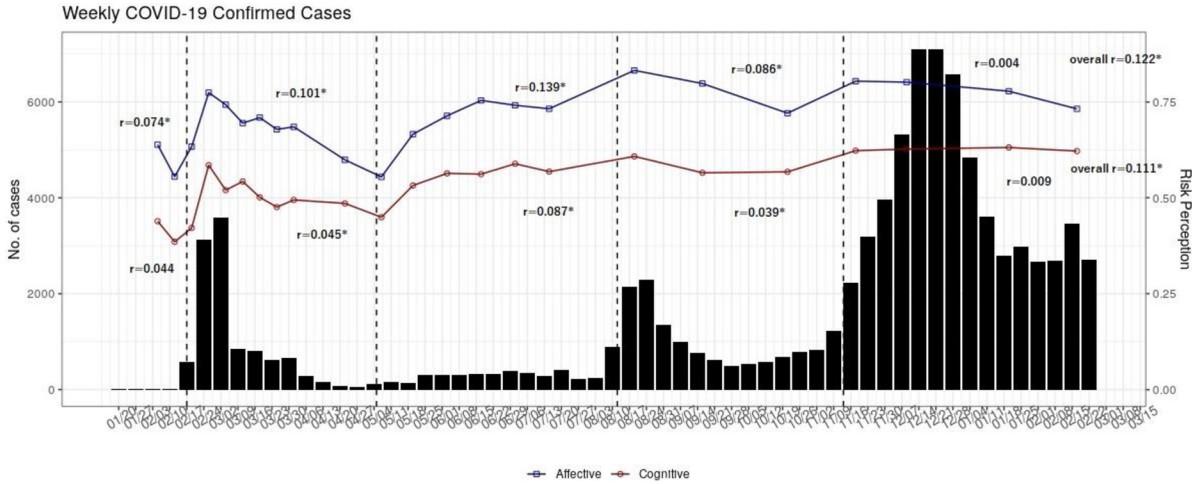

**Fig 1. Number of confirmed cases and cognitive and affective risk perceptions by phase.**

cluster outbreak (Phase 1), the first cluster outbreak started from the religious facilities in non-metropolitan areas (Phase 2), subsided intermediate periods (Phase 3), the second cluster outbreak started due to massive anti-government rallies in the metropolitan area (Phase 4), and the third cluster outbreak in which the coronavirus spread from nursing homes and healthcare facilities in the metropolitan area to an unspecified majority (Phase 5) (Fig 1, S2 Table in S1 File). The characteristics of the social events in each phase could be related to the perceived risk of COVID-19 and other associated factors. For example, in Phase 4, when the coronavirus outbreak was spread to metropolitan areas due to large-scale, anti-government protests led by far-right groups, people who trusted the government and the president (Democratic Party) and those with a liberal orientation may have experienced increased risk and threat perception compared to the conservatives.

Even though both dimensions of risk perception of COVID-19 may be related to trust in the government, political ideologies, and socio-demographic variables over time, relevant research is limited. This study investigated the association between these factors and risk perception (affective and cognitive), using a year-long, five-phase, longitudinal design to answer the following research questions (RQ).

RQ1: How did the associations between trust in the current government, political ideologies, socio-demographic factors, and risk perception vary depending on affective risk perception (ARP) and cognitive risk perception (CRP)?

RQ2: How did the associations between trust in the government, political ideologies, socio-demographic factors, and risk perception change across the five phases of the COVID-19 outbreak?

## Methods

### Study design and population

This study consisted of 23 independent and consecutive telephone surveys conducted over one year, from the first week of February 2020, when COVID-19 reportedly began in South Korea, to the third week of February 2021. Each survey was conducted by trained interviewers via computer-assisted telephone interviews (CATI; 85% of interviews on mobile phones and 15%

of interviews on landlines) at approximately two-week intervals. For each survey, 1,000 people (approximately) over the age of 18, across the country, were randomly selected from a digit-dialing sample frame (S1 Table in S1 File). Stratified samples were extracted by age, gender, and region, and weights were assigned proportionally to the parameters for each stratification to ensure that the participants were representative of all South Korean adults. In total, 23,018 individuals participated in the survey and were included in the analysis. All the surveys were conducted by Gallup Korea, an affiliate of Gallup International. Such telephone survey-based consecutive cross-sectional studies have been considered during other critical infectious disease outbreaks to monitor community response, starting from the initial phase of the epidemic [27, 47]. Detailed information about the study population by phase and survey is presented in S3 and S4 Tables of S1 File.

## Measures

Factors potentially associated with both the risk perception dimensions included socio-demographic factors, trust in the government, and political ideologies.

**Demographics.** Socio-demographic factors included gender, age, occupation, household economic status, and region of residence. Age (in years) was divided into 18–29, 30–39, 40–49, 50–59, and 60 and above [9, 48]. Occupations were classified as unemployed, farming/forestry/fishing, self-employed, blue-collar workers, white-collar workers, full-time homemaker and students. Self-reported household economic status was classified as lower/lower middle, middle, upper middle/upper. The residential area was divided into five regions (Metropolitan area, Seoul, Incheon, and Gyeonggi Province; Chungcheong, a middle region; Yeongnam, a south-eastern region; Honam, a south-western region; None of the above, Gangwon/Jeju Province) [48, 49].

**Political characteristics.** Trust in the current government was evaluated by asking, "Do you approve or disapprove of the way President Moon Jae-in is handling his job?" Respondents were asked to choose one of the following options–"approval," "disapproval," "neither/nor," and "do not know." Political ideologies were measured as "conservative", "liberal", "neutral", and "no opinion".

**Risk perception.** The ARP was rated on a four-point scale using the question, "How worried are you about contracting the COVID-19 infection?" The answers were rated as "very worried" (4), "a little worried" (3), "not very worried" (2), and "not worried at all" (1). The CRP was assessed on a four-point scale, using the question, "How likely are you to contract COVID-19?" The answers were rated as "very likely" (4), "somewhat likely" (3), "not very likely" (2), and "not at all likely" (1). Owing to the urgency of the outbreak, the validity of the questionnaires was not assessed. For a simple analysis and results, the scores for both risk perceptions were reclassified as dichotomous. Scores 1 and 2 were combined to indicate "not a perceived risk," and scores 3 and 4 were combined to indicate "perceived risk." Shentu and his colleagues [50] noted that dichotomization might reduce the estimation bias when the response was contaminated by measurement errors. However, if the measurement was accurate, dichotomization implied a loss of information that could lead to conservative results [51].

## Analysis

The one year was divided into five phases, as defined by the Korean Disease Control and Prevention Agency and the South Korean government, based on the number of confirmed cases and specific social events (S2 Table in S1 File). In addition to the factors mentioned above, the number of confirmed cases in the country (log scale) was included as a potentially associated factor, as in the previous studies [10]. One can expect the risk perceptions in South Korea to

vary according to the number of confirmed cases. Since almost all South Korean adults have access to daily updates and news via the internet and mass media, these numbers are regularly reported to the public [52]. This was considered in the previous studies [10], whereby, intuitively, risk perceptions varied according to the number of confirmed cases.

We reported the survey response rates over time. The relationship between each factor and risk perception was investigated by univariate analyses, using the chi-squared test (categorical variables) and two-sample t-test (numeric variables). The correlations between the number of confirmed cases and the two dimensions of risk perception were evaluated using Pearson's correlation coefficient and a t-test for correlation. Multiple logistic regression models were used to evaluate the adjusted odds ratio (aOR) and the confidence interval (CI) for the associated effect of each factor on the two dimensions. Eight covariates were considered explanatory variables to control for each other's effects. To observe the change in the effect of these factors by phase, an analysis was performed separately for each phase. The researchers calculated the $p$-values that tested the homogeneity of the aORs over each phase for each factor ($p$-value for trend), using likelihood ratio tests. Respondents with any missing values were lower than 2.9% of the study population and excluded from the analysis.

## Ethics

This study was reviewed and approved by the Institutional Review Board (IRB) of the Seoul Metropolitan Government-Seoul National University Boramae Medical Center (IRB No. 07-2021-38). The need for informed consent was waived by the IRB because the data were analyzed anonymously.

## Results

### Descriptive statistics and time trends of the ARP and the CRP

Table 1 presents that the overall proportion of the people who perceived affective and cognitive risk were 71.4% and 53.6%, respectively. Both risk perceptions were significantly different based on trust in the current government, political ideologies, and socio-demographic factors (p <0.05). The ARPs were higher than the CRPs in all the subgroups for all the associated factors. However, the ARP and the CRP levels did not maintain a proportional pattern across the subgroups. For example, those in their 60s and above had the highest ARP, yet the lowest CRP. In addition, women had higher ARP yet lower CRP than men.

The ARPs were higher than the CRPs in all five phases (Fig 1). In each phase, the ARPs rose and fell faster than the CRPs. Overall, there were subtle significant positive correlations between the number of confirmed cases and the ARP and the CRP (r = 0.1). However, the phase-wise correlations with the CRP were smaller than those with the ARP. Although the highest number of confirmed cases occurred in Phase 5; the level of risk perception did not increase.

### Pooled analysis

Fig 2 presents the magnitude and direction of the associations between the eight factors and the two dimensions. Most of the factors differed in strength and direction of associations with risk perception. The ARP for individuals aged ≥ 30 years was significantly greater than the baseline group (18–29 years), although it was unclear whether the ARP increased with age. Conversely, the CRP was significantly lower for individuals aged ≥ 30 years than the baseline group (18–29 years) and it decreased with increasing age. Women had significantly higher ARP than men; however, there was no significant gender difference in the CRP levels. Although there was no significant region-wise difference in the ARP, non-metropolitan

**Table 1. Overall levels of affective and cognitive risk perceptions.**

| | Overall | | Risk Perception of COVID-19 | | | | | |
|---|---|---|---|---|---|---|---|---|
| | Respondents | | Affective | | | Cognitive | | |
| | N | (%) | N | (%) | *p*-value | N | (%) | *p*-value |
| Total | 23,018 | 100% | 16,434 | 71.4% | | 12,213 | 53.6% | |
| **Age (years)** | | | | | | | | |
| Mean ± SD | 49.3 ± 16.7 | | 49.9 ± 16.9 | | < 0.001 | 46.2 ± 15.8 | | < 0.001 |
| 18–29 | 3,630 | 15.8% | 2,466 | 67.9% | < 0.001 | 2,280 | 62.8% | < 0.001 |
| 30–39 | 3,505 | 15.2% | 2,496 | 71.2% | | 2,188 | 62.4% | |
| 40–49 | 4,397 | 19.1% | 3,003 | 68.3% | | 2,613 | 59.4% | |
| 50–59 | 4,802 | 20.9% | 3,381 | 70.4% | | 2,498 | 52.0% | |
| 60+ | 6,684 | 29.0% | 4,997 | 74.8% | | 2,634 | 39.4% | |
| **Gender** | | | | | < 0.001 | | | 0.005 |
| Men | 11,613 | 50.5% | 7,797 | 67.1% | | 6,364 | 54.8% | |
| Women | 11,405 | 49.5% | 8,546 | 74.9% | | 5,849 | 51.3% | |
| **Job** | | | | | < 0.001 | | | < 0.001 |
| Unemployed | 2,722 | 11.8% | 2,041 | 75.0% | | 1,222 | 44.9% | |
| Farming/Forestry/Fishing | 682 | 3.0% | 491 | 72.0% | | 244 | 35.8% | |
| Self-employed | 3,339 | 14.5% | 2,319 | 69.5% | | 1,797 | 53.8% | |
| Blue-collar | 3,552 | 15.4% | 2,448 | 68.9 | | 1,902 | 53.5% | |
| White-collar | 7,051 | 30.6% | 4,869 | 69.1% | | 4,307 | 61.1% | |
| Homemaker and Student | 5,579 | 24.2% | 4,175 | 74.8% | | 2,741 | 49.1% | |
| **Self-reported Household Economic Status** | | | | | < 0.001 | | | < 0.001 |
| Upper/Upper Middle | 3,617 | 15.7% | 2,398 | 66.3% | | 2,056 | 56.8% | |
| Middle | 10,331 | 44.9% | 7,248 | 70.2% | | 5,564 | 53.9% | |
| Lower Middle/Lower | 9,070 | 39.4% | 6,697 | 73.8% | | 4,593 | 50.6% | |
| **Residential Area** | | | | | 0.001 | | | < 0.001 |
| Metropolitan | 11,561 | 50.2% | 8,250 | 71.4% | | 6,488 | 56.1% | |
| Chungcheong | 2,384 | 10.4% | 1,709 | 71.7% | | 1,246 | 52.3% | |
| Yeongnam | 2,270 | 9.9% | 1,539 | 67.8% | | 1,009 | 44.4% | |
| Honam | 5,807 | 25.2% | 4,168 | 71.8% | | 2,986 | 51.4% | |
| None of the above | 996 | 4.3% | 677 | 68.0% | | 484 | 48.6% | |
| **Trust in the Current Government** | | | | | < 0.001 | | | < 0.001 |
| Approval | 11,368 | 49.4% | 7,415 | 65.2% | | 5,736 | 50.5% | |
| Disapproval | 9,668 | 42.0% | 7,507 | 77.6% | | 5,568 | 57.6% | |
| Neither/Nor | 857 | 3.7% | 616 | 71.9% | | 409 | 47.7% | |
| No Opinion | 1,125 | 4.9% | 805 | 71.6% | | 500 | 44.4% | |
| **Political Ideology** | | | | | < 0.001 | | | < 0.001 |
| Conservative | 5,822 | 25.3% | 4,333 | 74.4% | | 3,142 | 54.0% | |
| Liberal | 6,568 | 28.5% | 4,761 | 64.4% | | 3,567 | 57.9% | |
| Neutral | 6,670 | 29.0% | 4,294 | 72.5% | | 3,861 | 54.3% | |
| No opinion | 3,958 | 17.2% | 2,955 | 74.7% | | 1,643 | 41.5% | |

Notes: *p*-values were calculated using the chi-square test (categorical variables) and the two-sample t-test (numeric variables).

regions had a significantly lower CRP level than metropolitan regions. Similarly, the occupation had an insignificant effect on the ARP, except in the self-employed group; however, the CRP levels in the self-employed, blue-collar, and white-collar groups were significantly higher than in the unemployed group.

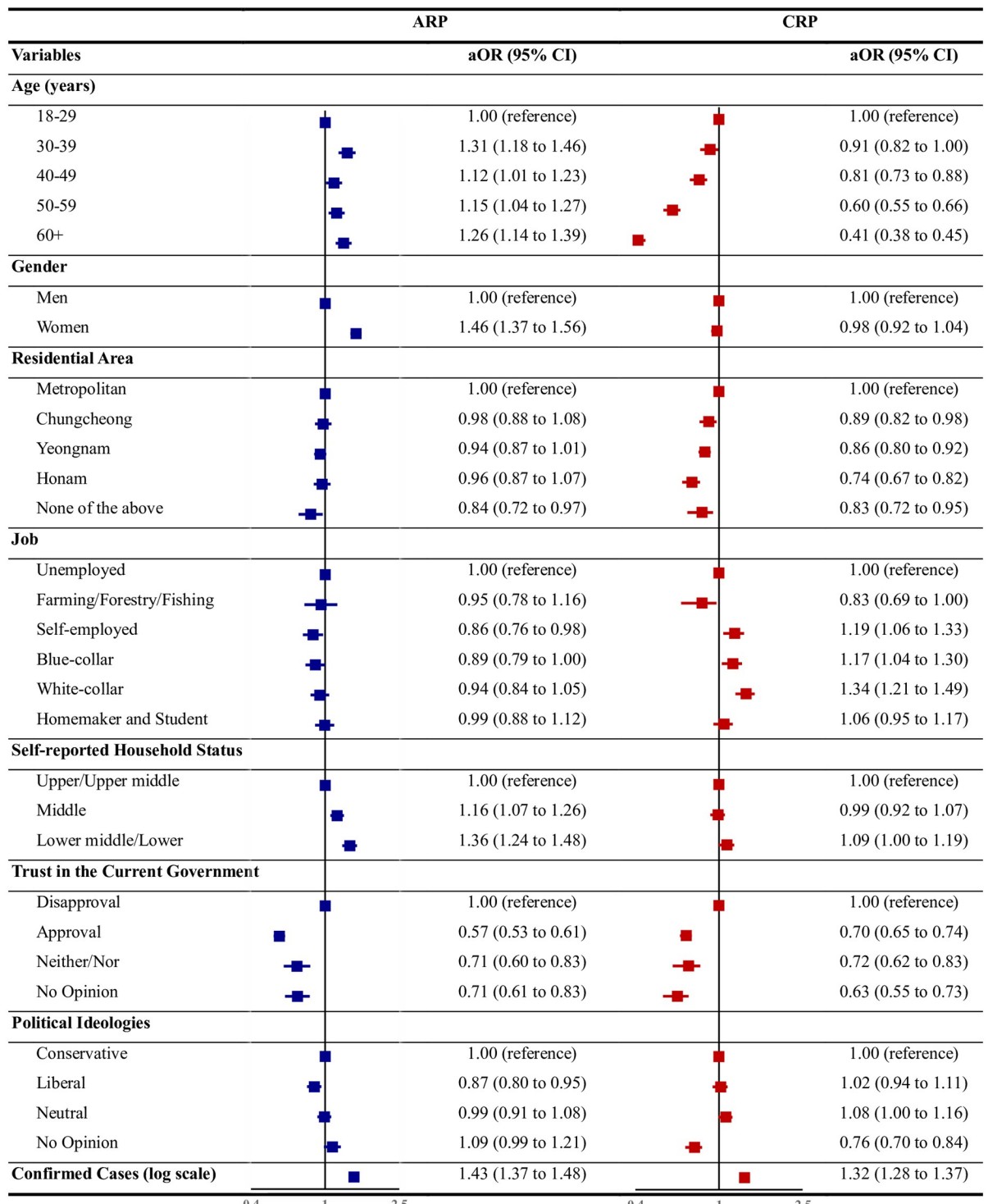

| | ARP | | CRP |
|---|---|---|---|
| **Variables** | **aOR (95% CI)** | | **aOR (95% CI)** |
| **Age (years)** | | | |
| 18-29 | 1.00 (reference) | | 1.00 (reference) |
| 30-39 | 1.31 (1.18 to 1.46) | | 0.91 (0.82 to 1.00) |
| 40-49 | 1.12 (1.01 to 1.23) | | 0.81 (0.73 to 0.88) |
| 50-59 | 1.15 (1.04 to 1.27) | | 0.60 (0.55 to 0.66) |
| 60+ | 1.26 (1.14 to 1.39) | | 0.41 (0.38 to 0.45) |
| **Gender** | | | |
| Men | 1.00 (reference) | | 1.00 (reference) |
| Women | 1.46 (1.37 to 1.56) | | 0.98 (0.92 to 1.04) |
| **Residential Area** | | | |
| Metropolitan | 1.00 (reference) | | 1.00 (reference) |
| Chungcheong | 0.98 (0.88 to 1.08) | | 0.89 (0.82 to 0.98) |
| Yeongnam | 0.94 (0.87 to 1.01) | | 0.86 (0.80 to 0.92) |
| Honam | 0.96 (0.87 to 1.07) | | 0.74 (0.67 to 0.82) |
| None of the above | 0.84 (0.72 to 0.97) | | 0.83 (0.72 to 0.95) |
| **Job** | | | |
| Unemployed | 1.00 (reference) | | 1.00 (reference) |
| Farming/Forestry/Fishing | 0.95 (0.78 to 1.16) | | 0.83 (0.69 to 1.00) |
| Self-employed | 0.86 (0.76 to 0.98) | | 1.19 (1.06 to 1.33) |
| Blue-collar | 0.89 (0.79 to 1.00) | | 1.17 (1.04 to 1.30) |
| White-collar | 0.94 (0.84 to 1.05) | | 1.34 (1.21 to 1.49) |
| Homemaker and Student | 0.99 (0.88 to 1.12) | | 1.06 (0.95 to 1.17) |
| **Self-reported Household Status** | | | |
| Upper/Upper middle | 1.00 (reference) | | 1.00 (reference) |
| Middle | 1.16 (1.07 to 1.26) | | 0.99 (0.92 to 1.07) |
| Lower middle/Lower | 1.36 (1.24 to 1.48) | | 1.09 (1.00 to 1.19) |
| **Trust in the Current Government** | | | |
| Disapproval | 1.00 (reference) | | 1.00 (reference) |
| Approval | 0.57 (0.53 to 0.61) | | 0.70 (0.65 to 0.74) |
| Neither/Nor | 0.71 (0.60 to 0.83) | | 0.72 (0.62 to 0.83) |
| No Opinion | 0.71 (0.61 to 0.83) | | 0.63 (0.55 to 0.73) |
| **Political Ideologies** | | | |
| Conservative | 1.00 (reference) | | 1.00 (reference) |
| Liberal | 0.87 (0.80 to 0.95) | | 1.02 (0.94 to 1.11) |
| Neutral | 0.99 (0.91 to 1.08) | | 1.08 (1.00 to 1.16) |
| No Opinion | 1.09 (0.99 to 1.21) | | 0.76 (0.70 to 0.84) |
| **Confirmed Cases (log scale)** | 1.43 (1.37 to 1.48) | | 1.32 (1.28 to 1.37) |

**Fig 2. Associations between related factors and cognitive and affective risk perceptions.** Notes: Affective risk perception (ARP), cognitive risk perception (CRP), adjusted odds ratio (aOR). The reported aORs are the exponentials of the fitted coefficients of the logistic regression models; the midpoint of each CI is 1.

The lower income group had significantly increased ARP compared to the upper and upper middle income groups; however, there were no significant differences in the CRP levels. Trust in the current government was the only factor with the same direction of association between the two risk perceptions, with significantly low values in all the groups. Political ideologies had an insignificant association with the dimensions of risk perception. The number of confirmed cases showed a significant positive relationship with both the ARP and the CRP levels; however, indicating a slightly stronger relationship with the former.

## Stratified analysis (over phase)

Tables 2 and 3 represent the changes in the associated effects when the analysis was stratified over time (phase) for the ARP and the CRP, respectively. The results of the pooled analysis (Fig 2) were reproduced without significant changes in all the phases; however, women's ARP changed over time (*p*-value for trend = 0.0378). This may be because the ARP in women was lower in Phase 4 than in the other phases. The factors that showed a significant change in their effect on the ARP were trust in the current government (p for trend < 0.0001), no trust in the current government (*p*-value for trend = 0.0200), liberal political orientation (*p*-value for trend = 0.0033), and the number of confirmed cases (*p*-value for trend < 0.0001). Individuals who indicated that they trusted the current government, who were politically liberal, or had no opinion had a stronger ARP in Phase 4 than in the other phases. The effect of the number of confirmed cases on the ARP was significant during Phases 1 through 4; however, it was insignificant in Phase 5 (Table 2).

The association of these factors with the CRP by phase was largely reproduced in the pooled analysis results. However, there were statistically significant changes in the effects of age, residential area, trust in the current government, and political ideologies. For example, aOR value for the impact of trust in the current government on the CRP was significantly below 1 in Phases 1–3 and 5; however, it was the highest in Phase 4 (aOR = 0.85, 95% CI, 0.71, 1.02). Similarly, the politically liberal group had the highest aOR in Phase 4 (aOR = 1.34, 95% CI, 1.07, 1.69). The number of confirmed cases had a significant effect on the CRP in Phases 1–3; however, it had little effect in Phases 4 and 5 (Table 3).

## Discussion

This study investigated how the two dimensions of risk perceptions—affective and cognitive—are related to trust in the current government, political ideologies, and socio-demographic factors and how they evolved with the progress of the coronavirus pandemic. This study revealed that the ARP and the CRP address different dimensions of risk perception by delineating their distinct associations with their respective factors. The results of the associations between the CRP and its related factors strongly indicate that the risk perception is based on the condition of the environment of the perceiver. For example, outdoor activities decrease with increasing age; however, there is no difference in the biological vulnerability of men and women to COVID-19. The number of confirmed cases is concentrated in the metropolitan area, and the likelihood of infection was higher when working in an indoor office environment. Hence, these findings indicate that the CRP is based on rational thinking to a certain extent and the general public continues to think rationally about the likelihood of infection even during an active pandemic [53].

Trust in the current government was the only factor that presented an association in the same direction for both the ARP and the CRP. People who did not trust the government were more concerned about contracting COVID-19 and considered themselves more likely to contract the disease than those who trusted the current government or had no clear political

**Table 2. Factors associated with affective risk perception of COVID-19 infection by phase.**

| | Phase 1 | Phase 2 | Phase 3 | Phase 4 | Phase 5 | |
|---|---|---|---|---|---|---|
| Variables | aOR (95% CI) | aOR (95% CI) | aOR (95% CI) | aOR (95% CI) | aOR (95% CI) | *p*-value for trend |
| **Age (years)** | | | | | | |
| 18–29 | 1.00 (reference) | 1.00 (reference) | 1.00 (reference) | 1.00 (reference) | 1.00 (reference) | |
| 30–39 | 1.20 (0.86 to 1.68) | 1.53 (1.29 to 1.83) | 1.15 (0.94 to 1.41) | 1.11 (0.80 to 1.54) | 1.34 (1.01 to 1.77) | 0.2163 |
| 40–49 | 0.97 (0.70 to 1.33) | 1.18 (1.01 to 1.40) | 1.13 (0.93 to 1.37) | 0.92 (0.68 to 1.26) | 1.11 (0.86 to 1.44) | 0.6888 |
| 50–59 | 0.83 (0.60 to 1.14) | 1.36 (1.15 to 1.60) | 1.16 (0.95 to 1.40) | 0.88 (0.65 to 1.19) | 1.12 (0.86 to 1.45) | 0.0533 |
| 60+ | 1.11 (0.81 to 1.52) | 1.45 (1.23 to 1.70) | 1.29 (1.07 to 1.55) | 0.92 (0.68 to 1.23) | 1.14 (0.89 to 1.47) | 0.0955 |
| **Gender** | | | | | | |
| Men | 1.00 (reference) | 1.00 (reference) | 1.00 (reference) | 1.00 (reference) | 1.00 (reference) | |
| Women | 1.65 (1.33 to 2.04) | 1.46 (1.31 to 1.63) | 1.53 (1.35 to 1.74) | 1.28 (1.05 to 1.56) | 1.51 (1.27 to 1.79) | 0.0378 |
| **Residential Area** | | | | | | |
| Metropolitan | 1.00 (reference) | 1.00 (reference) | 1.00 (reference) | 1.00 (reference) | 1.00 (reference) | |
| Chungcheong | 1.00 (0.72 to 1.38) | 1.00 (0.84 to 1.19) | 0.91 (0.75 to 1.11) | 1.00 (0.74 to 1.37) | 0.97 (0.75 to 1.26) | 0.9689 |
| Yeongnam | 0.92 (0.73 to 1.17) | 1.03 (0.91 to 1.17) | 0.88 (0.76 to 1.01) | 0.88 (0.71 to 1.10) | 0.96 (0.79 to 1.16) | 0.5072 |
| Honam | 0.96 (0.69 to 1.34) | 0.90 (0.76 to 1.07) | 0.88 (0.72 to 1.07) | 0.95 (0.69 to 1.31) | 1.34 (1.00 to 1.78) | 0.3898 |
| None of the above | 0.78 (0.48 to 1.26) | 0.79 (0.61 to 1.01) | 0.91 (0.68 to 1.22) | 0.76 (0.50 to 1.18) | 0.87 (0.59 to 1.28) | 0.8735 |
| **Job** | | | | | | |
| Unemployed | 1.00 (reference) | 1.00 (reference) | 1.00 (reference) | 1.00 (reference) | 1.00 (reference) | |
| Farming/Forestry/Fishing | 1.29 (0.66 to 2.51) | 0.93 (0.65 to 1.32) | 0.67 (0.46 to 0.97) | 1.13 (0.63 to 2.04) | 1.25 (0.72 to 2.17) | 0.2707 |
| Self-employed | 0.91 (0.60 to 1.36) | 0.85 (0.69 to 1.06) | 0.89 (0.70 to 1.13) | 0.78 (0.54 to 1.11) | 0.93 (0.68 to 1.27) | 0.9578 |
| Blue-collar | 1.05 (0.72 to 1.55) | 0.88 (0.71 to 1.08) | 0.85 (0.67 to 1.07) | 0.82 (0.58 to 1.16) | 0.97 (0.71 to 1.31) | 0.9149 |
| White-collar | 1.30 (0.89 to 1.89) | 0.95 (0.78 to 1.15) | 0.84 (0.67 to 1.05) | 1.05 (0.75 to 1.47) | 0.85 (0.64 to 1.13) | 0.399 |
| Homemaker and Student | 1.20 (0.82 to 1.76) | 0.92 (0.75 to 1.12) | 0.93 (0.74 to 1.17) | 1.11 (0.79 to 1.57) | 1.13 (0.84 to 1.53) | 0.5648 |
| **Self-reported Household Status** | | | | | | |
| Upper/Upper middle | 1.00 (reference) | 1.00 (reference) | 1.00 (reference) | 1.00 (reference) | 1.00 (reference) | |
| Middle | 0.87 (0.66 to 1.14) | 1.16 (1.01 to 1.34) | 1.21 (1.03 to 1.42) | 1.25 (0.97 to 1.62) | 1.32 (1.07 to 1.63) | 0.0023 |
| Lower middle/Lower | 0.91 (0.68 to 1.23) | 1.43 (1.22 to 1.67) | 1.45 (1.22 to 1.72) | 1.37 (1.05 to 1.79) | 1.45 (1.15 to 1.82) | 0.5139 |
| **Trust in the Current Government** | | | | | | |
| Disapproval | 1.00 (reference) | 1.00 (reference) | 1.00 (reference) | 1.00 (reference) | 1.00 (reference) | |
| Approval | 0.36 (0.29 to 0.45) | 0.46 (0.40 to 0.52) | 0.78 (0.67 to 0.90) | 0.88 (0.71 to 1.09) | 0.63 (0.53 to 0.75) | <0.0001 |
| Neither/Nor | 0.41 (0.23 to 0.73) | 0.60 (0.45 to 0.80) | 0.81 (0.60 to 1.10) | 0.80 (0.52 to 1.25) | 1.10 (0.69 to 1.74) | 0.02 |
| No Opinion | 0.90 (0.54 to 1.51) | 0.63 (0.48 to 0.84) | 0.75 (0.56 to 0.99) | 0.91 (0.58 to 1.42) | 0.65 (0.46 to 0.93) | 0.6946 |
| **Political Ideologies** | | | | | | |
| Conservative | 1.00 (reference) | 1.00 (reference) | 1.00 (reference) | 1.00 (reference) | 1.00 (reference) | |
| Liberal | 1.08 (0.81 to 1.43) | 0.67 (0.58 to 0.78) | 0.90 (0.76 to 1.06) | 1.34 (1.02 to 1.75) | 0.94 (0.74 to 1.19) | 0.0002 |
| Neutral | 1.06 (0.82 to 1.38) | 0.91 (0.79 to 1.05) | 0.97 (0.82 to 1.15) | 1.30 (1.01 to 1.66) | 0.94 (0.76 to 1.17) | 0.1787 |
| No Opinion | 1.74 (1.25 to 2.44) | 0.97 (0.81 to 1.16) | 1.11 (0.91 to 1.36) | 1.33 (1.00 to 1.78) | 0.85 (0.65 to 1.10) | 0.0033 |
| **Confirmed Cases (log scale)** | 2.47 (1.53 to 3.99) | 1.47 (1.33 to 1.63) | 4.06 (3.07 to 5.36) | 2.76 (1.82 to 4.20) | 0.90 (0.52 to 1.58) | <0.0001 |

Notes: Adjusted odds ratio (aOR), confidence interval (CI). The reported aORs are the exponentials of the fitted coefficients of the logistic regression models; the midpoint of each CI is 1. The *p*-values for testing the existence of a trend (*p*-value for trend) were calculated from the likelihood ratio test.

stance. Although trust in the current government is not related to an individual's actual likelihood of getting infected, it has shown a significant association with the CRP. Hence, the lack of trust in the current government interferes with risk judgement with logical reasoning and may be related to emotional responses such as fear, anxiety, or anger. Considering that people with less trust in the current government have a higher rate of vaccine hesitancy [9], this factor

**Table 3. Factors associated with cognitive risk perception of COVID-19 infection by phase.**

| Variables | Phase 1 aOR (95% CI) | Phase 2 aOR (95% CI) | Phase 3 aOR (95% CI) | Phase 4 aOR (95% CI) | Phase 5 aOR (95% CI) | p-value for trend |
|---|---|---|---|---|---|---|
| **Age (years)** | | | | | | |
| 18–29 | 1.00 (reference) | 1.00 (reference) | 1.00 (reference) | 1.00 (reference) | 1.00 (reference) | |
| 30–39 | 0.78 (0.57 to 1.08) | 0.87 (0.74 to 1.03) | 1.00 (0.82 to 1.21) | 1.00 (0.76 to 1.32) | 0.80 (0.62 to 1.04) | 0.4083 |
| 40–49 | 0.67 (0.49 to 0.92) | 0.73 (0.63 to 0.85) | 0.95 (0.79 to 1.14) | 0.94 (0.72 to 1.22) | 0.70 (0.55 to 0.90) | 0.0355 |
| 50–59 | 0.40 (0.29 to 0.55) | 0.58 (0.49 to 0.67) | 0.63 (0.52 to 0.75) | 0.82 (0.63 to 1.06) | 0.54 (0.43 to 0.69) | 0.0001 |
| 60+ | 0.36 (0.26 to 0.49) | 0.39 (0.33 to 0.45) | 0.48 (0.41 to 0.57) | 0.50 (0.39 to 0.64) | 0.30 (0.24 to 0.38) | <0.0001 |
| **Gender** | | | | | | |
| Men | 1.00 (reference) | 1.00 (reference) | 1.00 (reference) | 1.00 (reference) | 1.00 (reference) | |
| Women | 0.97 (0.79 to 1.20) | 0.98 (0.89 to 1.09) | 1.06 (0.94 to 1.19) | 0.93 (0.79 to 1.09) | 0.90 (0.77 to 1.05) | 0.5238 |
| **Residential Area** | | | | | | |
| Metropolitan | 1.00 (reference) | 1.00 (reference) | 1.00 (reference) | 1.00 (reference) | 1.00 (reference) | |
| Chungcheong | 0.78 (0.57 to 1.08) | 0.97 (0.83 to 1.13) | 0.81 (0.67 to 0.97) | 0.93 (0.72 to 1.20) | 0.89 (0.71 to 1.13) | 0.4507 |
| Yeongnam | 0.75 (0.59 to 0.95) | 0.97 (0.87 to 1.09) | 0.75 (0.66 to 0.86) | 0.92 (0.76 to 1.11) | 0.82 (0.70 to 0.97) | 0.0138 |
| Honam | 0.87 (0.62 to 1.23) | 0.73 (0.62 to 0.86) | 0.72 (0.59 to 0.87) | 0.67 (0.51 to 0.88) | 0.81 (0.64 to 1.03) | 0.0009 |
| None of the above | 1.26 (0.78 to 2.03) | 0.83 (0.66 to 1.05) | 0.77 (0.59 to 1.01) | 0.76 (0.52 to 1.11) | 0.84 (0.60 to 1.19) | 0.6943 |
| **Job** | | | | | | |
| Unemployed | 1.00 (reference) | 1.00 (reference) | 1.00 (reference) | 1.00 (reference) | 1.00 (reference) | |
| Farming / Forestry / Fishing | 1.10 (0.56 to 2.16) | 0.74 (0.53 to 1.03) | 0.72 (0.50 to 1.04) | 1.04 (0.64 to 1.69) | 0.88 (0.57 to 1.36) | 0.6857 |
| Self-Employed | 1.21 (0.80 to 1.84) | 1.31 (1.08 to 1.59) | 1.08 (0.87 to 1.35) | 1.07 (0.79 to 1.45) | 1.34 (1.02 to 1.75) | 0.5473 |
| Blue-collar | 0.95 (0.64 to 1.41) | 1.23 (1.01 to 1.49) | 1.23 (1.00 to 1.52) | 1.13 (0.84 to 1.52) | 1.17 (0.90 to 1.51) | 0.9337 |
| White-collar | 1.37 (0.94 to 2.00) | 1.34 (1.12 to 1.61) | 1.39 (1.13 to 1.70) | 1.28 (0.96 to 1.70) | 1.51 (1.17 to 1.94) | 0.8283 |
| Homemaker and Student | 1.21 (0.82 to 1.78) | 1.01 (0.84 to 1.21) | 1.03 (0.84 to 1.26) | 1.08 (0.81 to 1.43) | 1.21 (0.94 to 1.55) | 0.7262 |
| **Self-reported Household Status** | | | | | | |
| Upper/Upper middle | 1.00 (reference) | 1.00 (reference) | 1.00 (reference) | 1.00 (reference) | 1.00 (reference) | |
| Middle | 0.99 (0.76 to 1.30) | 0.92 (0.80 to 1.05) | 1.04 (0.89 to 1.21) | 1.07 (0.86 to 1.33) | 1.09 (0.90 to 1.32) | 0.0237 |
| Lower middle/Lower | 0.88 (0.66 to 1.18) | 1.04 (0.90 to 1.21) | 1.16 (0.99 to 1.37) | 1.03 (0.81 to 1.30) | 1.30 (1.05 to 1.59) | 0.2846 |
| **Trust in the Current Government** | | | | | | |
| Disapproval | 1.00 (reference) | 1.00 (reference) | 1.00 (reference) | 1.00 (reference) | 1.00 (reference) | |
| Approval | 0.50 (0.40 to 0.63) | 0.61 (0.55 to 0.69) | 0.77 (0.68 to 0.88) | 0.85 (0.71 to 1.02) | 0.79 (0.67 to 0.93) | < 0.0001 |
| Neither/Nor | 0.67 (0.37 to 1.20) | 0.50 (0.38 to 0.65) | 0.66 (0.50 to 0.87) | 1.13 (0.77 to 1.66) | 1.15 (0.79 to 1.68) | 0.0007 |
| No opinion | 0.53 (0.33 to 0.85) | 0.52 (0.40 to 0.67) | 0.86 (0.66 to 1.11) | 0.72 (0.50 to 1.04) | 0.52 (0.38 to 0.72) | 0.0153 |
| **Political Ideologies** | | | | | | |
| Conservative | 1.00 (reference) | 1.00 (reference) | 1.00 (reference) | 1.00 (reference) | 1.00 (reference) | |
| Liberal | 0.90 (0.68 to 1.19) | 0.88 (0.77 to 1.01) | 1.11 (0.94 to 1.30) | 1.34 (1.07 to 1.69) | 1.02 (0.83 to 1.25) | 0.0198 |
| Neutral | 1.04 (0.81 to 1.35) | 0.99 (0.87 to 1.13) | 1.09 (0.93 to 1.26) | 1.20 (0.97 to 1.48) | 1.13 (0.94 to 1.36) | 0.5934 |
| No opinion | 0.79 (0.58 to 1.09) | 0.77 (0.66 to 0.90) | 0.75 (0.63 to 0.90) | 0.84 (0.66 to 1.07) | 0.67 (0.54 to 0.84) | 0.7028 |
| **Confirmed Cases (log scale)** | 1.95 (1.22 to 3.14) | 1.14 (1.04 to 1.25) | 2.36 (1.81 to 3.08) | 1.37 (0.98 to 1.92) | 0.97 (0.59 to 1.58) | < 0.0001 |

Note: Adjusted odds ratio (aOR), confidence interval (CI). The reported aORs are the exponentials of the fitted coefficients of the logistic regression models; the midpoint of each CI is 1. The p-values for testing the existence of a trend (p-value for trend) were calculated from the likelihood ratio test.

is related more to an emotional response than to rational thinking. The relationship between political ideologies and risk perception requires further exploration and interpretation within the South Korean political context. Previous studies have shown that people with a conservative political orientation generally have a lower risk perception; however, this study showed the opposite result for the ARP. In addition, the findings contrast with the results of a study in

which Conservative Party supporters, when it was the ruling party, indicated lower risk perception levels during the Middle East respiratory syndrome coronavirus (MERS-CoV) outbreak [27]. Thus, the political support for the ruling party, rather than an absolute political ideology, decisively correlates with the risk perception levels.

Although the overall magnitude or association direction of the related factors did not change substantially for each phase, minor changes were observed in Phase 4. This is probably related to the fact that Phase 4 was triggered by and spread through mass gatherings led by far-right groups who strongly opposed the ruling democratic government. Understandably, the ARP and the CRP levels of those who trust the current government or have the same political affiliation as the ruling party in 2020 increased in Phase 4. This indicates that political ideologies and trust in the current government are strongly associated with instinctive anxiety among the public.

Both the ARP and the CRP were not significantly associated with the number of confirmed cases in Phase 5. In fact, the CRP showed this pattern even in Phase 4. The number of confirmed cases increased; however, the actual risk perception did not increase. This suggests that the government's strong social distancing policy may not have been as impactful as the increase in the number of confirmed cases. This is in line with reports stating that social distancing is less effective in deterring people's movement as the COVID-19 pandemic becomes a prolonged event [54].

Regarding other important associated factors, gender plays a vital role in shaping risk perceptions. It is usually understood that women's risk perception level is higher than that of men; a view supported by the results obtained in this study. In addition, it was found that women have an increased ARP [39]. Furthermore, the study found that older people have a lower risk perception, particularly the CRP, which, too, is in line with previous studies [55, 56]. Previously, older age has been associated with less distress after the 9/11 attacks, with reduced fear of future attacks, and a steep decline in post-traumatic stress over time [57]. The present findings suggest that older adults have less risk perception about the COVID-19 crisis [37].

This study had several limitations. A major caveat is that this study did not include education level due to the limited information of the representative national survey for this study. Previous studies found associations between low levels of education and higher perceived severity and lower perceived probability [45]. Education level must be included in future research as it may affect the ability to acquire, comprehend, and communicate objective knowledge, which could predict reduced risk perception [15]. In addition, the factors that were important in other studies, such as direct experience, socio-cultural factors, psychological factors, trust in science, and media exposure, were not included in this analysis because they were beyond the scope of this study. However, future research must assess how media exposure or its use is related to personal risk perception levels and the mechanism of its relationships. When people socially experience risk, media functions as an "amplification station" for the social experience of risk by intensifying or weakening risk perception through its risk portrayal [58]. Indeed, fear and anger mediate the associations between social media exposure and MERS-CoV risk perception in South Korea [59]. Furthermore, although researchers assessed two major dimensions of risk perception during public health emergencies, each measure relied on a single item; thus, researchers could not verify their validity and reliability. This limitation was shared by the other variables in this study. Finally, this study did not include a disease severity measure, which may add to the perceived threat of COVID-19. However, as the fatality rate of COVID-19 in South Korea was relatively low compared to other epidemics, such as MERS-CoV, it may not be significantly associated with the major findings.

Notwithstanding its limitations, this study provides insights into how different risk perceptions were associated with trust in the current government, political ideologies, and socio-

demographic factors during the COVID-19 outbreak in South Korea. Our findings confirmed the empirical distinction between affective and cognitive risk perceptions concerning these factors. However, trust in the current government showed a correlation in the same direction for both dimensions—those with a lower level of trust in the current government, exhibited a higher level of risk perception. Although these results did not change significantly during the one-year observation period, they were associated with significant political events. Our results suggest that trust in the current government may play a role in shaping the risk perceptions of a pandemic, with potentially significant socio-demographic factors for public health outcomes. Risk perceptions are influenced by the changes in the underlying risk along with political interpretations of the risk. Therefore, it is necessary to design health risk communication messages, tailored for each target population group, to address the difference in the risk perception of COVID-19 according to socio-demographic backgrounds and political opinions. Indeed, a better understanding of risk perception and the socio-demographic and political factors linked with the perceived risk could help governments and health authorities implement countermeasures and policies in response to future public health emergencies.

## Supporting information

**S1 File.**
(DOCX)

## Author Contributions

**Conceptualization:** Yo Han Lee, Hyun-Hee Heo, Hyerim Noh, Deok Hyun Jang, Young-Geun Choi, Won Mo Jang, Jin Yong Lee.

**Data curation:** Deok Hyun Jang.

**Formal analysis:** Hyun-Hee Heo, Hyerim Noh, Deok Hyun Jang, Young-Geun Choi, Won Mo Jang.

**Investigation:** Deok Hyun Jang.

**Methodology:** Yo Han Lee, Hyun-Hee Heo, Hyerim Noh, Young-Geun Choi, Won Mo Jang.

**Project administration:** Won Mo Jang.

**Supervision:** Young-Geun Choi, Won Mo Jang.

**Validation:** Yo Han Lee, Won Mo Jang.

**Visualization:** Young-Geun Choi.

**Writing – original draft:** Yo Han Lee, Hyerim Noh.

**Writing – review & editing:** Yo Han Lee, Hyun-Hee Heo, Deok Hyun Jang, Young-Geun Choi, Won Mo Jang, Jin Yong Lee.

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
