## [Decision Letter · Decision Letter 0]

25 Jul 2022

PONE-D-22-09993Patterns and trends in factors associated with affective and cognitive risk perceptions of COVID-19PLOS ONE

Dear Dr. Jang,

Thank you for submitting your manuscript to PLOS ONE. After careful consideration, we feel that it has merit but does not fully meet PLOS ONE’s publication criteria as it currently stands. Therefore, we invite you to submit a revised version of the manuscript that addresses the points raised during the review process. Although the reviewers and I found your research question interesting, a number of concerns were raised. The reviewers have provided detailed comments, and I will not reiterate their points. However, an overarching issue is the lack of detail and explanation. With regard to the introduction, a more extensive review of the differences between affective and cognitive risk perception is necessary. What do these constructs differentially predict? Why is it important to examine ARP and CRP separately? Why might this matter in the context of COVID-19? A clear rationale for the study is lacking.  For the treatment of variables (e.g., dichotomizing) and analyses, a more detailed description of the analytic strategy should be provided. Also, please provide references supporting your analytic approach.  In the results section, it is unclear at times whether descriptions of the data are based on statistical comparisons or observation of the data. Be sure to report all statistical analyses.    In the limitations, all of your variables of interest (CRP, ARP, trust in government, political orientation) are measured with single items. This is a shortcoming that needs more attention. This study is also correlational. so causal claims cannot be made. In a few places, causal language is used to discuss the results. This is inappropriate and should be corrected.           

We look forward to receiving your revised manuscript.

Kind regards,

Natalie J. Shook

Academic Editor

PLOS ONE

Journal Requirements:

Jang DH is affiliated with Gallup Korea (https://www.gallup.co.kr/), but did not receive any funding from them for this work.

Reviewers' comments:

Reviewer's Responses to Questions

**Comments to the Author**

1. Is the manuscript technically sound, and do the data support the conclusions?

Reviewer #1: Partly

Reviewer #2: Yes

2. Has the statistical analysis been performed appropriately and rigorously? 

Reviewer #1: Yes

Reviewer #2: Yes

3. Have the authors made all data underlying the findings in their manuscript fully available?

Reviewer #1: No

Reviewer #2: No

4. Is the manuscript presented in an intelligible fashion and written in standard English?

Reviewer #1: No

Reviewer #2: Yes

5. Review Comments to the Author

Reviewer #1: Dear authors,

Authors have that they have analyze the trends and patterns in associations of two risk perceptions dimensions for COVID-19 —cognitive risk perception and affective risk perception— and their associated factors with a year-long five

phase longitudinal design. I have few suggestions to offer. The following are the recommendations that needs attention for this manuscript.

Title:

"Patterns and trends in factors associated with affective and cognitive risk perceptions of COVID-19", what is meant by pattern and trends, I mean to say, how these two are different from each other. Please explain both in context.

Abstract:

Be specific in your writing, try to reduce more text in abstract, and use numbers to claim your contributions.

Introduction:

The Introduction part is badly written. There is no logical flow and almost same things are mentioned repeatedly in haphazard manner. I would suggest the authors improve the introduction section of the article. I will recommend to re-write the introcudtion part and the content should a follow a logical flow.

Methods:

I am just curious to know if you also take into considertaion about the accessibility of the participants to the daily updates, news, internet. If I missed somewhere please highlight that or else at least discuss that in limitation.

My major concern with this study is the sample size and the methods used (lack of spatial analysis for spatial variation). South Korea is a big country with over 51 million people, therefore a sample size of less 23000 for spatial analysis is not reasonable. More importantly, the authors should provide information on the number of participants in each province everytime, like the the survey was conducted 23 times over a period of one year.

Results

A lot of over interpretation of the results is the major limitation.

Conclusion:

The conclusion is vague and didn't provide any clear and useful information . It needs to be rewritten

Reviewer #2: This study examined the associated predictors of risk perceptions during COVID-19. A multi-faceted approach was used to examine risk taking and a large sample was utilized. The study has some potential for publication but it’s not ready in the current form. More elaboration in the intro and clarification regarding the methodology is needed. I present my detailed comments below in the order of appearance in the text.

Abstract. “associated factors” is a vague term. There should be at least some reasoning for why these factors are associated. I comment more on this in the introduction.

p5 l78. “10 Asian, American, and European” what is the distribution of these 10 countries according to continents.

p5 l85. What is the justification for choosing a factor as “key”? You should elaborate more on why you choose specific factors and why you expected them to be associated. If there is a theoretical basis of these choices it should also be explained. Currently, the study is atheoretical and there is no solid background for why these factors are studied.

p5 l88. Do you expect significant differences between the five-phases? If yes, how? If not, what may this exploration show us in the end?

p5 l89. You’ve presented the details for the five phases in Table S2. I think these phases are an essential part of the study and possibly unfamiliar to non-Korean readers. Adding the necessary details in text would be helpful.

p5 l92. “related factors” as said this is vague term and I don’t see any reasoning for why only the factors that are present in the parentheses are chosen. These should have been introduced with justifications earlier.

p6 l101. This info on current standing of the literature should be presented earlier in the introduction.

p6 l115. So each survey actually has different samples, which makes over 23,000 people in total. From the abstract my first impression was that 23,000 people participated in 23 waves of the study. This should be clearly explained in the abstract.

Further, this is the first time I’ve encountered a design like this. For me to evaluate the following versions of the manuscript better I’d appreciate if the authors can include some example papers. Including these examples in text may also be helpful for the reader since this may not be common for many readers as well.

p7 l119. Please add the info that the descriptive information can be found at Table S3.

p7 l121. Instead of “potentially associated factors” it can be better to divide this as “demographics” and “political characteristics”. Then list them under a “measures” title along with the measured of “Risk perception”.

p7 l125. Why was age divided this way?

p7 l131. Why were these demographics chosen? For example, why is education not included?

p8 l142. Why were these scores combined and not used as a continuous measure?

p8 l145. This essential info should be presented in the introduction.

p8 l148. Are these confirmed cases based on the location of the participants or the whole country?

p9 l164. Are these percentages of people who perceived risk?

p15 l183. Add the indication that these are pooled analysis in the first place you start explaining the analysis.

p15 l 185. If the results are not statistically significant i don’t think they should be mentioned. They may raise more confusion.

p15 l188. I do not have expertise on this type of analysis and a question. You report that women’s ARP change over time according to the p-value of the test of homogeneity of aORs. How do you define that this is due to Phase 4 being lower than other phases? Did you test for differences between phases or is this just an interpretation based on the aORs?

Also for your confidence intervals you mid-point is not 0. This is also something I’m not used to. I'm not sure if this is a typo or a way of reporting that I'm used to. If it is a typo it should be fixed, if not it would be great to add in notes for what the mid point of the intervals are.

p19 l223. The other way around may also be present. Strong emotional responses may be provoking distrust towards the government. Did you test this possible other direction?

p21 l259. How the listed limitation actually limited should be explained. For example not including education level is listed as a limitation, but you did not explain how did this actually limited your study.

6. PLOS authors have the option to publish the peer review history of their article (what does this mean?). If published, this will include your full peer review and any attached files.

Reviewer #1: **Yes: **Dr Junaid Ahmad

Reviewer #2: **Yes: **Barış Sevi

---

## [Author Response · Author response to Decision Letter 0]

7 Oct 2022

Response to peer reviewer comments

Dear Natalie J. Shook,

Thank you for giving us the opportunity to submit a revised draft of our manuscript titled “The Association among Risk Perceptions of COVID-19, Trust in Government, Political Ideology, and Socio-Demographic Factors: A Year Consecutive Cross-Sectional Study in South Korea” [PONE-D-22-09993] to the PLOS ONE. We appreciate the time and effort that you and the reviewers have dedicated to providing valuable feedback on the manuscript. We are also grateful to the reviewers for their insightful comments on the paper. We have been able to incorporate changes in response to a majority of the suggestions provided by the reviewers. 

Below is a point-by-point response to the reviewers’ comments and concerns. 

Comments from Editorial Corrections

Comment 1: With regard to the introduction, a more extensive review of the differences between affective and cognitive risk perception is necessary. What do these constructs differentially predict? Why is it important to examine ARP and CRP separately? Why might this matter in the context of COVID-19? A clear rationale for the study is lacking.

- Response to the reviewer’s comment:

Thank you for your valuable comments. In response, we have removed vague expressions and rewritten the whole logical flow of the introduction section. (p4-7, 57-128)

Comment 2: For the treatment of variables (e.g., dichotomizing) and analyses, a more detailed description of the analytic strategy should be provided. Also, please provide references supporting your analytic approach.

- Response to the reviewer’s comment:

Thank you for bringing this important point. Due to the urgency of the outbreak, the validity of questionnaires on risk perception and government trust had not been assessed. Thus, we chose dichotomization for simple analysis and results. Shentu and his colleagues noted that dichotomization may reduce bias in estimation when the response is contaminated by measurement errors. However, if the measurement is accurate, the dichotomization implies loss of information that can lead to conservative results [MacCallum et al.]. We added this paragraph into the methods section(p9, 170-173).

Furthermore, we have updated the weakness of variables treatment in the limitation paragraph as follows:

“Third, although the researchers assessed two major dimensions of risk perception during public health emergencies, each measure relied on a single item, and thus the researchers could not verify their validity and reliability. It was also the same limitation to other variables of this study.” (p22-23, 284-287).

Comment 3: In the results section, it is unclear at times whether descriptions of the data are based on statistical comparisons or observation of the data. Be sure to report all statistical analyses.

- Response to the reviewer’s comment:

Thank you for your valuable comments. In response, we have updated the vague expressions in the results section according to your comments. (p10, 198-214)

Comment 4: In the limitations, all of your variables of interest (CRP, ARP, trust in government, political orientation) are measured with single items. This is a shortcoming that needs more attention. This study is also correlational. so causal claims cannot be made. In a few places, causal language is used to discuss the results. This is inappropriate and should be corrected.

- Response to the reviewer’s comment:

Thank you for your valuable comments. In response, we have removed the causality expression, and added the limitation of single item.

“Third, although the researchers assessed two major dimensions of risk perception during public health emergencies, each measure relied on a single item, and thus the researchers could not verify their validity and reliability. It was also the same limitation to other variables of this study.” (p22-23, 284-287).

 

Comments from Reviewer 1

Title & Abstract

1. Do the title and abstract cover the main aspect of the work? "Patterns and trends in factors associated with affective and cognitive risk perceptions of COVID-19", what is meant by pattern and trends, I mean to say, how these two are different from each other. Please explain both in context.

- Response to the reviewer’s comment:

Thank you for your valuable comments. In response, we have updated the title and the abstract section. (p3, 38-53)

Updated title: 

The Association among Risk Perceptions of COVID-19, Trust in Government, Political Ideology, and Socio-Demographic Factors: A Year Consecutive Cross-Sectional Study in South Korea.

2. Be specific in your writing, try to reduce more text in abstract, and use numbers to claim your contributions.

- Response to the reviewer’s comment:

Thank you for your valuable comments. In response, we have removed vague expressions and updated the abstract section. (p3, 38-53)

Introduction

3. The Introduction part is badly written. There is no logical flow and almost same things are mentioned repeatedly in haphazard manner. I would suggest the authors improve the introduction section of the article. I will recommend to re- write the introcudtion part and the content should a follow a logical flow.

- Response to the reviewer’s comment:

Thank you for your valuable comments. In response, we have removed vague expressions and rewritten the whole logical flow of the introduction section. (p4-7, 57-128)

Methods

4. I am just curious to know if you also take into considertaion about the accessibility of the participants to the daily updates, news, internet. If I missed somewhere please highlight that or else at least discuss that in limitation.

- Response to the reviewer’s comment:

Thank you for your valuable comments. In The accessibility of the participants to the daily updates, news, and internet was not included in the analysis. However, it is reasonable to assume that and almost all South Korean adults has the access to the daily updates and news via internet and mass media; for example, more than 95% of South Korean adults owned smartphone as of 2019. We addressed this point as follows:

“the number of confirmed cases in the whole country (log scaled) was also included as a potentially associated factor as in previous studies [10]. One can expect that risk perceptions in South Korea may tend to vary according to the number of confirmed cases. Because it is regularly reported to the public and almost all South Korean adults has the access to the daily updates and news via internet and mass media [50].” (p9, 178-182)

Furthermore, a review of the literature on the relationship between risk perception and media exposure was added to the introduction. Media exposure-related variables are beyond the scope of this study and are therefore mentioned as limitations in the discussion section.

“Risk perceptions often relate to how media represent a threat of risk [33, 34]. Perceived susceptibility and severity of risk tend to be affected by the way media frame the issue of threat. A systematic review of news coverage related to the H1N1 pandemic outbreak reveals that the excessive media coverage, overemphasis of threat over protection, and news coverage with frightening tone and manner influence amplifying the perceived risk [35]. A meta-analysis of 47 studies also suggests that concern for COVID-19 infection associated with media exposure worldwide [36].” (p5, 88-95)

“Second, factors that were found to be important in other studies, such as direct experience, socio-cultural factors, psychological factors, trust in science, and media exposure were not included in the analysis since they are beyond the scope of this study. However, future research is needed to assess how media exposure or media use is related to personal-level risk perception and the mechanism of its relationships. According to the social amplification of risk framework, media functions as an “amplification station” for the social experience of risk by intensifying or attenuating risk perception through its portrayal of risk [56]. Indeed, social media exposure is linked to the two emotions, fear and anger, and these emotions mediate the associations between social media exposure and risk perception towards MERS-CoV in South Korea [57].” (p22, 275-284)

5. My major concern with this study is the sample size and the methods used (lack of spatial analysis for spatial variation). South Korea is a big country with over 51 million people, therefore a sample size of less 23000 for spatial analysis is not reasonable.

- Response to the reviewer’s comment:

We agree with the point that spatial variation could be better captured via spatial analysis methods. To address the spatial variation with the limited sample size, we had included a categorized version of the participants’ residential area into five regions: (Yeongnam, a south-eastern region; Honam, a southwestern region; Capital Metro, a Seoul metropolitan area (Seoul, Incheon, and Gyeonggi Province), Chungcheong, and Gangwon/Jeju). This categorization is common in other studies on cognition and decision-making about political issues in Korea.

Kang, W.; Bae, J.S. Regionalism and party system change at the sub-national level: The 2016 Korean National Assembly Elec-tion. J. Inter. Area Stud. 2018, 25, 93–112. 

Moon, W. Decomposition of Regional Voting in South Korea. Party Politi. 2005, 11, 579–599.

Park HK, Ham JH, Jang DH, Lee JY, Jang WM. Political Ideologies, Government Trust, and COVID-19 Vaccine Hesitancy in South Korea: A Cross-Sectional Survey. Int J Environ Res Public Health. 2021 Oct 12;18(20):10655. doi: 10.3390/ijerph182010655. 

Kim JH, Jang DH, Jang WM. Association Between Self-Rated Political Orientation and Attitude Toward the Cash Transfer Policy During the COVID-19 Pandemic: A Nationwide Cross-Sectional Survey Conducted in South Korea. Front Public Health. 2022 May 17;10:887201. doi: 10.3389/fpubh.2022.887201

6. More importantly, the authors should provide information on the number of participants in each province everytime, like the the survey was conducted 23 times over a period of one year.

- Response to the reviewer’s comment:

Thank you for your constructive feedback. We updated Table S3 in the supplementary material to report the number of participants in each residential area and other demographic information for each of all 23 surveys. 

Results

7. A lot of over interpretation of the results is the major limitation.

- Response to the reviewer’s comment:

Thank you for your comments. We have removed the expression of over interpretation.

Conclusion

8. The conclusion is vague and didn't provide any clear and useful information. It needs to be rewritten

- Response to the reviewer’s comment:

Thank you for your valuable comments. In response, we revised the conclusion as follows.

“Notwithstanding its limitations, the present study provided insights into how two different risk perceptions associate with trust in government, political ideology, and sociodemographic factors during the COVID-19 outbreak in South Korea. Our findings confirmed the empirical distinction between affective and cognitive risk perceptions in relation with those factors. However, trust in the government showed a correlation in the same direction for both dimensions of risk perception; those with the lower level of trust in the government exhibited the higher level of risk perception. Although these results did not change significantly during the one-year observation period, they were associated with political events. Our results suggest that trust in government may play a role in shaping risk perceptions of a pandemic, with potentially significant socio-demographic factors for public health outcomes. Risk perceptions are influenced not only by changes in fundamental underlying risk, but also by political-related interpretations of the risk. Therefore, it is required to design health risk communication messages tailoring for each target population in a situation where the risk perception of COVID-19 differs according to various socio-demographic backgrounds and political opinions on the risk. Indeed, a better understanding of not only risk perception but also the sociodemographic and political-related factors that linked with perceived risk of the pandemic could help government and health authorities implement countermeasures and policies in response to any future public health emergencies.” (p23, 291-308)

 

Comments from Reviewer 2

Abstract

1. “associated factors” is a vague term. There should be at least some reasoning for why these factors are associated. I comment more on this in the introduction.

- Response to the reviewer’s comment:

Thank you for your valuable comments. In response, we have updated the abstract according to your comments. 

“This study aimed to investigate the association among two dimensions of risk perceptions for COVID-19 (risk as feeling and analysis), trust in the government, political ideology, and sociodemographic factors.” (p3, 39-41)

2. From the abstract my first impression was that 23,000 people participated in 23 waves of the study. This should be clearly explained in the abstract.

- Response to the reviewer’s comment:

Thank you for your valuable comments. In response, we have updated the abstract according to your comments. 

“This study used a year-long repeated cross-sectional design, in which a South Korean national sample (n=23,018) participated in 23 consecutive telephone surveys from February 2020 to February 2021.” (p3, 41-44)

Introduction

3. “10 Asian, American, and European” what is the distribution of these 10 countries according to continents (p5, 78).

- Response to the reviewer’s comment:

Thank you for your valuable comments. In response, we have removed vague and redundant sentences and rewritten the whole logical flow of the introduction section. (p4-7, 57-128)

4. What is the justification for choosing a factor as “key”? (p5, 85).

- Response to the reviewer’s comment:

Thank you for your valuable comments. In response, we have removed vague expressions like “key factors” and rewritten the whole logical flow of the introduction section. (p4-7, 57-128)

5. Do you expect significant differences between the five-phases? If yes, how? If not, what may this exploration show us in the end (p5, 88)?

- Response to the reviewer’s comment:

Thank you for your valuable comments. In response, we explained characteristics of each phase and its expected difference in introduction and discussed the related findings in the end.

“The five phases can be described as following before the first cluster outbreak (phase 1), the first cluster outbreak started from religious facilities of non-metropolitan area (phase 2), subsided intermediate period (phase 3), the second cluster outbreak started from massive anti-government rallies in the metropolitan area (phase 4), the third cluster outbreak, which coronavirus spread again from nursing homes and healthcare facilities in the metropolitan area to an unspecified majority (phase 5) (Fig 1, Table S2). The characteristics of social events in each phase may be related to the perceived risk of COVID-19 and potentially associated factors. For example, in phase 4, when the coronavirus outbreak was spread to metropolitan area due to large-scale anti-government protests led by far-right groups, people who support presidential job (Democratic Party) and those with a liberal orientation may increase risk perception of threat compared to the conservatives.” (p6, 105-116)

“This is probably related to the fact that Phase 4 was triggered by and spread through mass gatherings led by far-right groups who strongly opposed the government which democratic party president led. It is understandable that the ARP and CRP levels of those who trust in the government or have the same political affiliation as the ruling party in 2020 have increased in Phase 4.” (p21, 248-252)

6. You’ve presented the details for the five phases in Table S2. I think these phases are an essential part of the study and possibly unfamiliar to non-Korean readers. Adding the necessary details in text would be helpful (p5, 89).

- Response to the reviewer’s comment:

Thank you for your valuable comments. In response, we explained the details in the introduction section. (p6, 105-116)

7. “related factors” as said this is vague term and I don’t see any reasoning for why only the factors that are present in the parentheses are chosen. These should have been introduced with justifications earlier (p5, 92).

- Response to the reviewer’s comment:

Thank you for your valuable comments. In response, we explained the details in the introduction section. (p5, 81-101)

8. This info on current standing of the literature should be presented earlier in the introduction (p6, 101).

- Response to the reviewer’s comment:

Thank you for your valuable comments. In response, we have rewritten the whole logical flow of the introduction section. (p4-7, 57-128)

Methods

9. I’d appreciate if the authors can include some example papers. Including these examples in text may also be helpful for the reader since this may not be common for many readers as well (p6, 115).

- Response to the reviewer’s comment:

Thank you very much for detailed comments. Thank you for your valuable comments. In response, we revised the abstract and methods as follows.

“This study used a year-long repeated cross-sectional design, in which a South Korean national sample (n=23,018) participated in 23 consecutive telephone surveys from February 2020 to February 2021.” (Abstract; p3, 41-44)

“This study consisted of 23 independent and consecutive telephone surveys conducted over a one-year period from the first week of February 2020, when COVID-19 reportedly began in South Korea, to the third week of February 2021.” (Methods; p7, 132-134)

In addition, we cited example papers in which the design is telephone survey-based consecutive cross-sectional study in other urgent infectious diseases outbreak (Methods; p7, 142-145).

10. Please add the info that the descriptive information can be found at Table S3 (p7, 119).

- Response to the reviewer’s comment:

Thank you. We added the information in methods (p7, 144-145). 

11. Instead of “potentially associated factors” it can be better to divide this as “demographics” and “political characteristics”. Then list them under a “measures” title along with the measured of “Risk perception” (p7, 121).

- Response to the reviewer’s comment:

Thank you for the constructive suggestion. We renamed the paragraphs as suggested; see “Measures” subsection on page 7-8.

12. Why was age divided this way? (p7, 125)

- Response to the reviewer’s comment:

We used this categorized age in the analysis to encourage interpretability and simplicity of results, while we allow nonlinear relationship between age and risk perception. And this categorization is frequently and practically used in some survey studies. We have cited the references.

Kang, W.; Bae, J.S. Regionalism and party system change at the sub-national level: The 2016 Korean National Assembly Elec-tion. J. Inter. Area Stud. 2018, 25, 93–112. 

Jang WM, Jang DH, Lee JY. Social Distancing and Transmission-reducing Practices during the 2019 Coronavirus Disease and 2015 Middle East Respiratory Syndrome Coronavirus Outbreaks in Korea. J Korean Med Sci. 2020 Jun;35(23):e220. https://doi.org/10.3346/jkms.2020.35.e220

Jang WM, Kim UN, Jang DH, Jung H, Cho S, Eun SJ, Lee JY. Influence of trust on two different risk perceptions as an affective and cognitive dimension during Middle East respiratory syndrome coronavirus (MERS-CoV) outbreak in South Korea: serial cross-sectional surveys. BMJ Open. 2020 Mar 4;10(3):e033026. doi: 10.1136/bmjopen-2019-033026.

13. Why were these demographics chosen? For example, why is education not included? (p7, 131)

- Response to the reviewer’s comment:

We appreciate raising the point. Due to the length limitations of telephone-based survey questionnaire, some questions related to demographics, such as education, was omitted. We further clarified this point in the Discussion section as follows.

“This study has several limitations. First, the major caveat concerns that this study does not include education level due to the limited information of representative national survey for this study. Previous studies found associations between low education level and higher perceived severity, and between low education level and lower perceived probability [45]. Education level should be included in future research as it may affect the ability to acquire, comprehend, and communicate objective knowledge on coronavirus which could predict reduced risk perception [15]. Second, factors that were found to be important in other studies, such as direct experience, socio-cultural factors, psychological factors, trust in science, and media exposure were not included in the analysis since they are beyond the scope of this study. However, future research is needed to assess how media exposure or media use is related to personal-level risk perception and the mechanism of its relationships. According to the social amplification of risk framework, media functions as an “amplification station” for the social experience of risk by intensifying or attenuating risk perception through its portrayal of risk [56]. Indeed, social media exposure is linked to the two emotions, fear and anger, and these emotions mediate the associations between social media exposure and risk perception towards MERS-CoV in South Korea [57].” (Discussion; p22, 269-284)

14. Why were these scores combined and not used as a continuous measure? (p8, 142)

- Response to the reviewer’s comment:

Thank you for bringing this important point. Due to the urgency of the outbreak, the validity of questionnaires on risk perception and government trust had not been assessed. Thus, we chose dichotomization for simple analysis and results. Shentu and his colleagues noted that dichotomization may reduce bias in estimation when the response is contaminated by measurement errors. However, if the measurement is accurate, the dichotomization implies loss of information that can lead to conservative results [MacCallum et al.]. We added this paragraph into the main body on page 9, lines 170-173.

15. This essential info should be presented in the introduction (p8, 145).

- Response to the reviewer’s comment:

Thank you for the suggestion. We added more information in the Introduction section as follows. Thank you.

“Until February 2021, a year pandemic period can be divided into five phases according to the upsurge of confirmed cases of COVID-19 in South Korea [46]. Three times the cluster outbreak had occurred related to religious facilities, large-scale downtown gatherings, nursing homes, and healthcare facilities from January 2020 to February 2021. The five phases can be described as following before the first cluster outbreak (phase 1), the first cluster outbreak started from religious facilities of non-metropolitan area (phase 2), subsided intermediate period (phase 3), the second cluster outbreak started from massive anti-government rallies in the metropolitan area (phase 4), the third cluster outbreak, which coronavirus spread again from nursing homes and healthcare facilities in the metropolitan area to an unspecified majority (phase 5) (Fig 1, Table S2). The characteristics of social events in each phase may be related to the perceived risk of COVID-19 and potentially associated factors. For example, in phase 4, when the coronavirus outbreak was spread to metropolitan area due to large-scale anti-government protests led by far-right groups, people who support presidential job (Democratic Party) and those with a liberal orientation may increase risk perception of threat compared to the conservatives.” (Introduction; p6, 102-116)

16. Are these confirmed cases based on the location of the participants or the whole country? (p8, 148)

- Response to the reviewer’s comment:

It is based on the whole country. We clarified this on page 9, lines 178. Thank you.

Results

17. Are these percentages of people who perceived risk? (p9, 164)

- Response to the reviewer’s comment:

Yes, these are. We clarified this on page 10, lines 199-200. Thank you.

18. Add the indication that these are pooled analysis in the first place you start explaining the analysis (p15, 183).

- Response to the reviewer’s comment:

We appreciate the suggestion. We included titles for each paragraph in the Result section in the revised manuscript.

19. If the results are not statistically significant i don’t think they should be mentioned. They may raise more confusion (p15, 185).

- Response to the reviewer’s comment:

Thank you for bringing this important point. To avoid confusion, we excluded those redundant statistically insignificant results. The revised paragraph is on page 16, lines 198-214.

20. You report that women’s ARP change over time according to the p-value of the test of homogeneity of aORs. How do you define that this is due to Phase 4 being lower than other phases? (p15, 188)

- Response to the reviewer’s comment:

We deeply thank you for this clarifying question. It is the interpretation based on aOR; we clarified this point on page 16, lines 186-189.

21. Also for your confidence intervals you mid-point is not 0. This is also something I’m not used to. I'm not sure if this is a typo or a way of reporting that I'm used to. If it is a typo it should be fixed, if not it would be great to add in notes for what the mid point of the intervals are (p15, 188).

- Response to the reviewer’s comment:

Thank you for raising an important point. In the multivariable logistic regression model, the aOR is defined by $\\exp(\\beta)$ where $\\beta$ is a fitted coefficient corresponding to each factor. Thus, the mid-point becomes 1; we clarified that the midpoint 1 in the note of each table (lines 181-182, 198-199, 203-204)

Discussion

22. The other way around may also be present. Strong emotional responses may be provoking distrust towards the government. Did you test this possible other direction? (p19 223)

- Response to the reviewer’s comment:

As this study explores the relationship between risk perception and trust in government, the causal relationship was not tested. Therefore, the sentence that may cause confusion in the reader has been modified as follows.

“Although trust in the government is not related to an individual’s actual likelihood of getting infected, it has been shown to have a significant association with CRP. This means that lack of trust in the government interfere with a risk judgement using logic reasoning and may be related to emotional responses such as fear, anxiety, or anger.” (p20, 232-235)

23. How the listed limitation actually limited should be explained. For example not including education level is listed as a limitation, but you did not explain how did this actually limited your study (p21, 259).

- Response to the reviewer’s comment:

Thank you for the suggestion. We have updated more information in the discussion section as follows. Thank you.

“This study has several limitations. First, the major caveat concerns that this study does not include education level due to the limited information of representative national survey for this study. Previous studies found associations between low education level and higher perceived severity, and between low education level and lower perceived probability [45]. Education level should be included in future research as it may affect the ability to acquire, comprehend, and communicate objective knowledge on coronavirus which could predict reduced risk perception [15].” (p22, 269-275)

 

Journal Requirements

2. participant informed consent

- Response to the comment:

We have updated the ethics section as follows. Thank you.

This study was reviewed and approved by the Institutional Review Board (IRB) of the Seoul Metropolitan Government-Seoul National University Boramae Medical Center (IRB No. 07-2021-38). The need for informed consent was waived by the IRB due to the fact that the data were analyzed anonymously.

3. Competing Interests section

- Response to the comment:

We have updated the competing interests section as follows. Thank you.

Jang DH is affiliated with Gallup Korea (https://www.gallup.co.kr/), but did not receive any funding from them for this. This does not alter our adherence to PLOS ONE policies on sharing data and materials.

4. Data Availability statement

- Response to the comment:

We have been confirmed that out data availability statement was acceptable under PLOS guidelines according to email from Adam Thompson on September 6.

---

## [Decision Letter · Decision Letter 1]

1 Dec 2022

PONE-D-22-09993R1The Association among Risk Perceptions of COVID-19, Trust in Government, Political Ideology, and Socio-Demographic Factors: A Year Consecutive Cross-Sectional Study in South Korea.PLOS ONE

Dear Dr. Jang,

Thank you for submitting your manuscript to PLOS ONE. After careful consideration, we feel that it has merit but does not fully meet PLOS ONE’s publication criteria as it currently stands. Therefore, we invite you to submit a revised version of the manuscript that addresses the points raised during the review process.

Please see Reviewer 2's comments.

We look forward to receiving your revised manuscript.

Kind regards,

Natalie J. Shook

Academic Editor

PLOS ONE

Journal Requirements:

Reviewers' comments:

Reviewer's Responses to Questions

**Comments to the Author**

1. If the authors have adequately addressed your comments raised in a previous round of review and you feel that this manuscript is now acceptable for publication, you may indicate that here to bypass the “Comments to the Author” section, enter your conflict of interest statement in the “Confidential to Editor” section, and submit your "Accept" recommendation.

Reviewer #1: All comments have been addressed

Reviewer #2: (No Response)

2. Is the manuscript technically sound, and do the data support the conclusions?

Reviewer #1: Yes

Reviewer #2: Yes

3. Has the statistical analysis been performed appropriately and rigorously? 

Reviewer #1: Yes

Reviewer #2: I Don't Know

4. Have the authors made all data underlying the findings in their manuscript fully available?

Reviewer #1: Yes

Reviewer #2: No

5. Is the manuscript presented in an intelligible fashion and written in standard English?

Reviewer #1: Yes

Reviewer #2: Yes

6. Review Comments to the Author

Reviewer #1: Thank you for revising the manuscript. I think the comments are addressed very well. Thank you for giving us the opportunity to review the revised draft of your manuscript titled “The Association among Risk Perceptions of COVID-19, Trust in Government, PoliticalIdeology, and Socio-Demographic Factors: A Year Consecutive Cross-Sectional Study

in South Korea”. I don't have any more commments.

Reviewer #2: I thank the authors for addressing my comments. The manuscript looks much improved. I have a few more comments taht I believe will help the manuscript improve even more.

P8L147. Not sure "Factors that were potentially associated with both dimensions of risk perception included sociodemographic factors, trust in the government, and political ideology." this sentence belong under the subtitle demographics. Maybe it should have come right after the Measures title.

P8L157. Since you are asking about the current president and not asking about general attitudes towards government it would be more appropriate to label this variable as "Trust in current government". This point may also be discussed in the discussion section.

Results. The analyses plan explains what analyses are conducted. However mentioning the name of the type of tests you conducted before reporting the results would be helpful for readers who are not custom to your analyses. For example you report a p value at p10l203, but it's not clear what test is used. You can also add this info at the tables, I think it would be very helpful for the readers.

Table 1. The label "Presidential job approval rating" is not consistent with how you labeled this variable in your measures.

Table 3&4. At table 2 you indicated the reference variables, this was not done at Tables 3 and 4. Is there a reason for this difference? If not I think it helps to include that indication.

Overall comment. Labels of gender (men, women) and sex (male, female) are used interchangeably. Please be consistent on the use of this variable.

7. PLOS authors have the option to publish the peer review history of their article (what does this mean?). If published, this will include your full peer review and any attached files.

Reviewer #1: **Yes: **Junaid Ahmad

Reviewer #2: No

---

## [Author Response · Author response to Decision Letter 1]

14 Dec 2022

Response to peer reviewer comments

Dear Natalie J. Shook,

Thank you for giving us the opportunity to submit a 2nd revised draft of our manuscript titled “The Association among Risk Perceptions of COVID-19, Trust in Government, Political Ideology, and Socio-Demographic Factors: A Year Consecutive Cross-Sectional Study in South Korea” [PONE-D-22-09993R1] to the PLOS ONE. We appreciate the time and effort that you and the reviewers have dedicated to providing valuable feedback on the manuscript. We are also grateful to the reviewers for their insightful comments on the paper. We have been able to incorporate changes in response to a majority of the suggestions provided by the reviewers. 

Below is a point-by-point response to the reviewers’ comments and concerns. 

Comments from Reviewer 2

Methods

1. Not sure "Factors that were potentially associated with both dimensions of risk perception included sociodemographic factors, trust in the government, and political ideology." this sentence belong under the subtitle demographics. Maybe it should have come right after the Measures title (P8L147).

- Response to the reviewer’s comment:

Thank you for your thorough review. Edited as suggested.

2. Since you are asking about the current president and not asking about general attitudes towards government it would be more appropriate to label this variable as "Trust in current government". This point may also be discussed in the discussion section (P8L157).

- Response to the reviewer’s comment:

We are grateful for the precise feedback. We updated P8L157 and the others accordingly.

Results

3. The analyses plan explains what analyses are conducted. However mentioning the name of the type of tests you conducted before reporting the results would be helpful for readers who are not custom to your analyses. For example you report a p value at p10l203, but it's not clear what test is used. You can also add this info at the tables, I think it would be very helpful for the readers.

- Response to the reviewer’s comment:

Thank you very much for important point. To convey accurate information to readers, we added the type of statistical test we used in each table. In addition, we further specified the type of tests and analysis for Table 1 and Figure 1 in the "Analysis" paragraph in the Method section, like the following:

"... We reported survey response rates over time. The relationships between each factor and risk perception were investigated by univariate analyses with the chi-squared test (categorical variables) and the two-sample t-test (numeric variables). The correlations between the number of confirmed cases and the two dimensions of risk perception were evaluated using Pearson’s correlation coefficient and the t-test for correlation. ...."

4. Table 1. The label "Presidential job approval rating" is not consistent with how you labeled this variable in your measures.

- Response to the reviewer’s comment:

We have updated all sentences from "presidential job approval rating" to "trust in the current government."

5. Table 3&4. At table 2 you indicated the reference variables, this was not done at Tables 3 and 4. Is there a reason for this difference? If not I think it helps to include that indication.

- Response to the reviewer’s comment:

Thank you for important point. We added indication "1.00 (reference)" into the reference groups in Tables 3&4.

6. Overall comment. Labels of gender (men, women) and sex (male, female) are used interchangeably. Please be consistent on the use of this variable.

- Response to the reviewer’s comment:

We are grateful for careful review. The updated version replaced sex (male, female) by the labels of gender (men, women) for consistency.

Lastly, We have added information about the support for YGC’s work (P23l374).

---

## [Editor Report · Decision Letter 2]

10 Jan 2023

The association between the risk perceptions of COVID-19, trust in government, political ideologies, and socio-demographic factors: A year-long cross-sectional study in South Korea.

PONE-D-22-09993R2

Dear Dr. Jang,

We’re pleased to inform you that your manuscript has been judged scientifically suitable for publication and will be formally accepted for publication once it meets all outstanding technical requirements.

Kind regards,

Natalie J. Shook

Academic Editor

PLOS ONE
---

## [Editor Report · Acceptance letter]

14 Mar 2023

PONE-D-22-09993R2 

The association between the risk perceptions of COVID-19, trust in the government, political ideologies, and socio-demographic factors: A year-long cross-sectional study in South Korea. 

Dear Dr. Jang:

I'm pleased to inform you that your manuscript has been deemed suitable for publication in PLOS ONE. Congratulations! Your manuscript is now with our production department. 

Kind regards, 

on behalf of

Dr. Natalie J. Shook 

Academic Editor

PLOS ONE